# Data-driven Distributionally Robust Polynomial Optimization

**Martin Mevissen**
IBM Research—Ireland
martmevi@ie.ibm.com

**Emanuele Ragnoli**
IBM Research—Ireland
eragnoli@ie.ibm.com

**Jia Yuan Yu**
IBM Research—Ireland
jy@osore.ca

## Abstract

We consider robust optimization for polynomial optimization problems where the uncertainty set is a set of candidate probability density functions. This set is a ball around a density function estimated from data samples, i.e., it is data-driven and random. Polynomial optimization problems are inherently hard due to nonconvex objectives and constraints. However, we show that by employing polynomial and histogram density estimates, we can introduce robustness with respect to distributional uncertainty sets without making the problem harder. We show that the optimum to the distributionally robust problem is the limit of a sequence of tractable semidefinite programming relaxations. We also give finite-sample consistency guarantees for the data-driven uncertainty sets. Finally, we apply our model and solution method in a water network optimization problem.

## 1 Introduction

For many optimization problems, the objective and constraint functions are not adequately modeled by linear or convex functions (*e.g.*, physical phenomena such as fluid or gas flow, energy conservation, etc.). Non-convex polynomial functions are needed to describe the model accurately. The resulting polynomial optimization problems are hard in general. Another salient feature of real-world problems is uncertainty in the parameters of the problem (*e.g.*, due to measurement errors, fundamental principles, or incomplete information), and the need for optimal solutions to be robust against worst case realizations of the uncertainty. Robust optimization and polynomial optimization are already an important topic in machine learning and operations research. In this paper, we combine the polynomial and uncertain features and consider robust polynomial optimization.

We introduce a new notion of *data-driven distributional* robustness: the uncertain problem parameter is a probability distribution from which samples can be observed. Consequently, it is natural to take as the uncertainty set a set of functions, such as a norm ball around an *estimated* probability distribution. This approach gives solutions that are less conservative than classical robust optimization with a set for the uncertain parameters. It is easy to see that the set uncertainty setting is an extreme case of a distributional uncertainty set comprised of a set of Dirac densities. This stands in sharp contrast with real-world problems where more information is at hand than the support of the distribution of the parameters affected by uncertainty. Uncertain parameters may follow normal, Poisson, or unknown nonparametric distributions. Such parameters arise in queueing theory, economics, etc.

We employ methods from both machine learning and optimization. First, we take care to estimate the distribution of the uncertain parameter using *polynomial* basis functions. This ensures that the resulting robust optimization problem can be reduced to a polynomial optimization problem. In turn, we can then employ an iterative method of SDP relaxations to solve it. Using tools from machine learning, we give a finite-sample consistency guarantee on the estimated uncertainty set. Using tools from optimization, we give an asymptotic guarantee on the solutions of the SDP relaxations.

Section 2 presents the model of data-driven distributionally robust polynomial optimization—DRO for short. Section 3 situates our work in the context of the literature. Our contributions are the following. In Section 4, we consider the general case of uncertain *multivariate* distribution, which yields a generalized problem of moments for the distributionally robust counterpart. In Section 5, we introduce an efficient histogram approximation for the case of uncertain *univariate* distributions, which yields instead a polynomial optimization problem for the distributionally robust counterpart. In Section 6, we present an application of our model and solution method in the domain of water network optimization with real data.

## 2 Problem statement

Consider the following polynomial optimization problem

$$\min_{x \in X} \quad h(x, \xi), \tag{1}$$

where $\xi \in \mathbb{R}^n$ is an uncertain parameter of the problem. We allow $h$ to be a polynomial in $x \in \mathbb{R}^m$ and $X$ to be a basic closed semialgebraic set. That is, even if $\xi$ is fixed, (1) is a hard problem in general.

In this work, we are interested in *distributionally robust optimization* (DRO) problems that take the form

$$(\text{DRO}) \quad \min_{x \in X} \max_{f \in \mathfrak{D}_{\varepsilon, N}} \quad \mathbb{E}_f \, h(x, \xi), \quad \text{for all } t, \tag{2}$$

where $x$ is the decision variable, $\xi$ is a random variable distributed according to an unknown probability density function $f^*$, which is the *uncertain parameter* in this setting. The expectation $\mathbb{E}_f$ is with respect to a density function $f$, which belongs to an *uncertainty set* $\mathfrak{D}_{\varepsilon, N}$. This uncertainty set itself is a set of possible *probability density functions* constructed from a given sequence of *samples* $\xi_1, \ldots, \xi_N$ distributed i.i.d. according to the unknown density function $f^*$ of the uncertain parameter $\xi$. We call $\mathfrak{D}_{\varepsilon, N}$ a *distributional uncertainty set*, it is a *random* set constructed as follows:

$$\mathfrak{D}_{\varepsilon, N} = \{f : \text{a prob. density s.t. } \|f - \widehat{f}_N\| \leqslant \varepsilon\}, \tag{3}$$

where $\varepsilon > 0$ is a given constant, $\|\cdot\|$ is a norm, and $\widehat{f}_N$ is an density function estimated from the samples $\xi_1, \ldots, \xi_N$. We describe the construction of the distributional uncertainty set in the cases of multivariate and univariate samples in Sections 4 and 5.

We say that a robust optimization problem is *data-driven* when the uncertainty set is an element of a sequence of uncertainty sets $\mathfrak{D}_{\varepsilon, 1} \supseteq \mathfrak{D}_{\varepsilon, 2} \supseteq \ldots$, where the index $N$ represents the number of samples of $\xi$ observed by the decision-maker. This definition allows us to completely separate the problem of robust optimization from that of constructing the appropriate uncertainty set $\mathfrak{D}_{\varepsilon, N}$. The underlying assumption is that the uncertainty set (due to finite-sample estimation of the parameter $\xi$) adapts continuously to the data as the sample size $N$ increases. By considering data-driven problems, we are essentially employing tools from statistical learning theory to derive consistency guarantees.

Let $\mathbb{R}[x]$ denote the vector space of real-valued, multivariate polynomials, *i.e.*, every $g \in \mathbb{R}[x]$ is a function $g : \mathbb{R}^m \to \mathbb{R}$ such that

$$g(x) = \sum_{|\alpha| \leqslant d} g_\alpha x^\alpha = \sum_{|\alpha| \leqslant d} g_\alpha x_1^{\alpha_1} \ldots x_m^{\alpha_m}, \, \alpha \in \mathbb{N}^m,$$

where $\{g_\alpha\}$ is a set of real numbers. A polynomial optimization problem (POP) is given by

$$\min_{x \in K} q(x), \tag{4}$$

where $K = \{x \in \mathbb{R}^d \mid g_1(x) \geqslant 0, \ldots, g_m(x) \geqslant 0\}$, $q \in \mathbb{R}[x]$, and $g_j \in \mathbb{R}[x]$ for $j = 1, \ldots, m$.

One of our key results arises from the observation that the distributional robust counterpart of a POP is a POP as well. A set $K$ defined by a finite number of multivariate polynomial inequality constraints is called a *basic closed semialgebraic set*. As shown in [1], if the basic closed semi-algebraic set $K$ compact and archimedian, there is a hierarchy of SDP relaxations whose minima

converge to the minimum of (4) for increasing order of the relaxation. Moreover, if (4) has an unique minimal solution $x^\star$, then the optimal solution $y_\tau^\star$ of the $\tau$-th order SDP relaxation converges to $x^\star$ as $\tau \to \infty$.

Our work combines robust optimization with notions from statistical machine learning, such as density estimation and consistency. Our data-driven robust polynomial optimization method applies to a number of machine learning problems. One example arises in Markov decision problems where a high-dimensional value-function is approximated by a low-dimensional polynomial $V$. A distributionally robust variant of value iteration can be cast as:

$$\max_{a \in A} \min_{f \in \mathcal{D}_{\varepsilon,N}} \mathbb{E}_f\{r(x,a,\xi) + \gamma \sum_{x' \in X} P(x' \mid x,a,\xi)V(x')\},$$

where $\xi$ is a random parameter with unknown distribution and the uncertainty set $\mathcal{D}_{\varepsilon,N}$ of possible distribution is constructed by estimation. We present next two further examples.

**Example 2.1** (Distributionally robust ridge regression)**.** We are given an i.i.d. sequence of observation-label samples $\{(\xi_i, y_i) \in \mathbb{R}^{n-1} \times \mathbb{R} : i = 1, \dots, N\}$ from an unknown distribution $f^*$, where each observation $\xi_i$ has an associated label $y_i \in \mathbb{R}$. Ridge regression minimizes the empirical residual with $\ell^2$-regularization and uses the samples to construct residual function. The distributionally robust version of ridge regression is a conceptually different approach: it uses the samples to construct a random uncertainty set $\mathfrak{D}_{\varepsilon,N}$ to estimate the distribution $f^*$ and can be formulated as

$$\min_{u \in \mathbb{R}^n} \max_{f \in \mathfrak{D}_{\varepsilon,N}} \quad \mathbb{E}_f(y_{N+1} - \xi_{N+1} \cdot u)^2 + \lambda(u \cdot u),$$

where $\mathfrak{D}_{\varepsilon,N}$ is the uncertainty set of possible densities constructed from the $N$ samples. Our solution methods can even be applied to regression problems with nonconvex loss and penalty functions.

**Example 2.2** (Robust investment)**.** Optimization problems of the form of (2) arise in problems that involve monetary measures of risk in finance [2]. For instance, the problem of robust investment in a vector of (random) financial positions $\xi \in \mathbb{R}^n$ is

$$\min_{v \in \Delta^n} \sup_{Q \in \mathcal{Q}} -\mathbb{E}_Q\big[U(v \cdot \xi)\big],$$

where $\mathcal{Q}$ denotes a set of probability distributions, $U$ is a utility function, and $v \cdot \xi$ is an allocation among financial positions. If $U$ is polynomial, then the robust utility functional is a special case of DRO.

## 3 Our contribution in context

To situate our work within the literature, it is important to note that we consider *distributional* uncertainty sets and *polynomial* constraints and objectives. In this section, we outline related works with different and similar uncertainty sets, constraints and objectives.

Robust optimization problems of the form of (2) have been studied in the literature with different uncertain sets. In several works, the uncertainty sets are defined in terms of moment constraints [3, 4, 5]. Moment based uncertainty sets are motivated by the fact that probabilistic constraints can be replaced by constraints on the first and second moments in some cases [6].

In contrast, we do not consider moment constraints, but distributional uncertainty sets based on probability density functions with the $L_p$-norm as the metric. One reason for our approach is that higher moments are difficult to estimate [7]. In contrast, probability density functions can be readily estimated using a variety of data-driven methods, *e.g.*, empirical histograms, kernel-based [8, 9], and orthogonal basis [10] estimates. Uncertainty sets defined by distribution-based constraints appear also in problems of risk measures [11]. For example uncertainty sets defined using Kantorovich distance are considered in [5, Section 4] and [11] while [5, Section 3] and [12] consider distributional uncertainty with both measure bounds (of the form $\mu_1 \leqslant \mu \leqslant \mu_2$) and moment constraints.

[13] considers distributional uncertainty sets with a $\phi$-divergence metric. A notion of distributional uncertainty set has also been studied in the setting of Markov decision problems [14]. However, in those works, the uncertainty set is not data-driven.

Robust optimization formulations for polynomial optimization problems have been studied in [1, 15] with deterministic uncertainty sets (*i.e.*, neither distributional, nor data-driven). A contribution is to show how to transform distributionally robust counterparts of polynomial optimization problems into polynomial optimization problems. In order to solve these POP, we take advantage of the hierarchy of SDP relaxations from [1]. Another contribution of this work is to use sampled information to construct distributional uncertainty sets more suitable for problems where more and more data is collected over time.

## 4   Multivariate uncertainty around polynomial density estimate

In this section, we construct a data-driven uncertainty set in the $L_2$-space—with the uniform norm $\|\cdot\|_2$. Furthermore we assume, the support of $\xi$ is contained in some basic closed semialgebraic set $S := \{z \in \mathbb{R}^n \mid s_j(z) \geqslant 0, \, j = 1, \ldots, r\}$, where $s_j \in \mathbb{R}[z]$.

In order to construct a data-driven distributional uncertainty set, we need to estimate the density $f^*$ of the parameter $\xi$. Various density estimation approaches exist—*e.g.*, kernel-density and histogram estimation. Some of these give rise to a computational problem due to the curse of dimensionality. However, to ensure that the resulting robust optimization problem remains an polynomial optimization problem, we define the empirical density estimate $\widehat{f}_N$ as a multivariate polynomial (cf. Section 2).

Let $\{\pi_k\}$ denote univariate Legendre polynomials:

$$\pi_k(a) = \sqrt{\frac{2k+1}{2}} \frac{1}{2^k k!} \frac{\mathrm{d}^k}{\mathrm{d}a^k}(a^2 - 1)^k, \quad a \in \mathbb{R}, k = 0, 1, \ldots$$

Let $\alpha \in \mathbb{N}^n$, $z \in \mathbb{R}^n$, and $\pi_\alpha(z) = \pi_{\alpha_1}(z_1) \ldots \pi_{\alpha_n}(z_n)$ denote the multivariate Legendre polynomial. In this section, we employ the following Legendre series density estimator [10]: $\widehat{f}_N(z) = \sum_{|\alpha| \leqslant d} \frac{1}{N} \sum_{j=1}^N \pi_\alpha(\xi_j) \pi_\alpha(z)$.

In turn, we define the following uncertainty set:

$$\mathfrak{D}_{d,\epsilon,N} = \left\{ f \in \mathbb{R}[z]_d \mid \int_S f(z) \, \mathrm{d}z = 1, \, \left\| f - \widehat{f}_N \right\|_2 \leqslant \epsilon \right\}.$$

where $\mathbb{R}[z]_d$ denotes the vector space of polynomials in $\mathbb{R}[z]$ of degree at most $d$. Observe that the polynomials in $\mathfrak{D}_{d,\epsilon,N}$ are not required to be non-negative on $S$. However, the non-negativity constraint on $S$ can be added at the expense of making the resulting DRO problem for a POP a generalized problem of moments.

### 4.1   Solving the DRO

Next, we present asymptotic guarantees for solving distributionally robust polynomial optimization through SDP relaxations. This result rests on the following assumptions, which are detailed in [1].

**Assumption 4.1.** The sets $X = \{x \in \mathbb{R}^m \mid k_j(z) \geqslant 0, \, j = 1, \ldots, t\}$ and $S = \{z \in \mathbb{R}^n \mid s_j(z) \geqslant 0, \, j = 1, \ldots, r\}$ are compact. There exist $u \in \mathbb{R}[x]$ and $v \in \mathbb{R}[z]$ such that $u = u_0 + \sum_{j=1}^t u_j \, k_j$ and $v = v_0 + \sum_{j=1}^r v_j \, s_j$ for some sum-of-squares polynomials $\{u_j\}_{j=0}^t$, $\{v_j\}_{j=0}^r$, and the level sets $\{x \mid u(x) \geqslant 0\}$ and $\{z \mid v(z) \geqslant 0\}$ compact.

Note that sets $X$ and $S$ satisfying Assumption 4.1 are called *archimedian*. This assumption is not much more restrictive than compactness, *e.g.*, if $S := \{z \in \mathbb{R}^n \mid s_j(z) \geqslant 0, \, j = 1, \ldots, r\}$ is compact, then there exists a $L^2$-ball of radius $R$ that contains $S$. Thus, $S = \tilde{S} = \{z \in \mathbb{R}^n \mid s_j(z) \geqslant 0, \, j = 1, \ldots, r, \sum_{i=1}^n z_i^2 \leqslant R\}$. With Theorem 1 in [22] it follows that $\tilde{S}$ satisfies Assumption 4.1.

**Theorem 4.1.** *Suppose that Assumption 4.1 holds. Let $h \in \mathbb{R}[x, z]$, $\widehat{f}_N \in \mathbb{R}[z]$, and let $X$ and $S$ be basic closed semialgebraic sets. Let $V^\star \in \mathbb{R}$ denote the optimum of problem*

$$\min_{x \in X} \max_{f \in \mathfrak{D}_{d,\varepsilon,N}} \int_S h(x, z) f(z) \mathrm{d}z. \tag{5}$$

(i) *Then, there exists a sequence of SDP relaxations $SDP_r$ such that $\min SDP_r \nearrow V^\star$ for $r \to \infty$.*

(ii) *If (5) has a unique minimizer $x^\star$, and $m_r$ the sequence of subvectors of optimal solutions of $SDP_r$ associated with the first order moments of monomials in $x$ only. Then, $m_r \to x^\star$ componentwise for $r \to \infty$.*

All proofs appear in the appendix of the supplementary material.

## 4.2 Consistency of the uncertainty set

In this section, we show that the uncertainty set that we constructed is consistent. In other words, given constants $\epsilon$ and $\delta$, we give number of samples $N$ needed to ensure that the closest polynomial to the unknown density $f^*$ belongs to the uncertainty set $\mathfrak{D}_{d,\varepsilon,N}$ with probability $1 - \delta$.

**Theorem 4.2** ([10, Section 3]). *Let $c_\alpha$ denote the coefficients $c_\alpha = \int \pi_\alpha f^*$ for all values of the multi-index $\alpha$. Suppose that the density function $f^*$ is square-integrable. We have $\mathbb{E}\|f^* - \widehat{f}_N\|_2^2 \leqslant C_H \sum_{\alpha:|\alpha|\leqslant d} \min(1/N, c_\alpha^2)$, where $C_H$ is a constant that depends only on $f^*$.*

As a corollary of Theorem 4.2, we obtain the following.

**Corollary 4.3.** *Suppose that the assumptions of Theorem 4.2 hold. Let $g_d^*$ denote the polynomial function $g_d^*(x) = \sum_{\alpha:|\alpha|\leqslant d} c_\alpha x^\alpha$. There exists a function[1] $\Phi$ such that $\Phi(d) \searrow 0$ as $d \to \infty$ and such that*

$$\mathbb{P}(g_d^* \in \mathfrak{D}_{d,\varepsilon,N}) \geqslant 1 - \frac{C_H \sum_{\alpha:|\alpha|\leqslant d} \min(1/N, c_\alpha^2) + \Phi^2(d)}{(\varepsilon - \Phi(d))^2},$$

*for $\varepsilon > \Phi(d)$.*

*Remark* 1. Observe that since $\sum_{\alpha:|\alpha|\leqslant d} \min(1/N, c_\alpha^2) \leqslant \binom{n+d}{d}/N = (n + d)!/(N\ d!\ n!)$, by an appropriate choice of $N$, it is possible to guarantee that the right-hand side tends to zero, even as $d \to \infty$.

## 5 Univariate uncertainty around histogram density estimate

In this section, we describe an additional layer of approximation for the univariate uncertainty setting. In contrast to Section 4, by approximating the uncertainty set $\mathfrak{D}_{\varepsilon,N}$ by a set of histogram density functions, we reduce the DRO problem to a polynomial optimization problem of degree identical with the original problem. Moreover, we derive finite-sample consistency guarantees. We assume that samples $\xi_1, \ldots, \xi_N$ are given for the uncertain parameter $\xi$, which takes values in a given interval $[A, B] \subset \mathbb{R}$. I.e., in contrast to the previous section, we assume that the uncertain parameter takes values in a bounded interval. We partition $\mathbb{R}$ into $K$-intervals $u_0, \ldots, u_{K-1}$, such that $|u_k| = |B - A|/K$ for all $k = 0, \ldots, K - 1$. Let $m_0, \ldots, m_{K-1}$ denote the midpoints of the respective intervals. We define the empirical density vector $\widehat{p}_{N,K}$:

$$\widehat{p}_{N,K}(k) = \frac{1}{N} \sum_{i=1}^N 1_{[\xi_i \in u_k]} \quad \text{for all } k = 0, \ldots, K - 1.$$

Recall that the $L_\infty$-norm of a function $G : X \to \mathbb{R}^n$ is: $\|G\|_\infty = \sup_{x \in X} |G(x)|$. In this section, we approximate the uncertainty set $\mathfrak{D}_{\varepsilon,N}$ by a subset of the simplex in $\mathbb{R}^K$:

$$\mathfrak{W}_{\varepsilon,N} = \left\{ p \in \Delta^K : \|p - \widehat{p}_{N,K}\|_\infty \leqslant \varepsilon \right\},$$

where $p = (p_1, \ldots, p_K)$ denote a vector in $\mathbb{R}^K$. In turn, this will allow us to approximate the DRO problem (2) by the following:

$$(\text{ADRO}): \quad \min_{x \in X} \max_{p \in \mathfrak{W}_{\varepsilon,N}} \sum_{k=0}^{K-1} h(x, m_k)\ p_k. \tag{6}$$

## 5.1 Solving the DRO

The following result is an analogue of Theorem 4.1.

**Theorem 5.1.** *Suppose that Assumption 4.1 holds. Let $h \in \mathbb{R}[x, z]$, and let $X$ be basic closed semialgebraic[2]. Let $W^\star \in \mathbb{R}$ denote the optimum of problem*

$$\min_{x \in X} \max_{p \in \mathfrak{W}_{\varepsilon, N}} \sum_{k=0}^{K-1} h\left(x, m_k\right) p_k. \tag{7}$$

   (i) *Then, there exists a sequence of SDP relaxations $SDP_r$ such that $\min SDP_r \nearrow W^\star$ for $r \to \infty$.*

   (ii) *If (7) has a unique minimizer $x^\star$, let $m_r$ the sequence of subvectors of optimal solutions of $SDP_r$ associated with the first order moments of the monomials in $x$ only. Then, $m_r \to x^\star$ componentwise for $r \to \infty$.*

## 5.2 Approximation error

Next, we bound the error of approximating $\mathfrak{D}_{\varepsilon, N}$ with $\mathfrak{W}_{\varepsilon, N}$. This error depends only on the "degree" $K$ of the histogram approximation.

**Theorem 5.2.** *Suppose that the support of $\xi$ is the interval $[A, B]$. Suppose that $|h(x, z)| \leqslant H$ for all $x \in X$ and $z \in [A, B]$. Let $\tilde{M} \triangleq \sup\{f''(z) : f \in \mathfrak{D}_{\gamma, N}, z \in [A, B]\}$ be finite. Let $g_x(z) \triangleq h(x, z)f(z)$ and let $M \triangleq \sup\{g'_x(z) : f \in \mathfrak{D}_{\gamma, N}, z \in [A, B]\}$ be finite. For every $\gamma \leqslant K\varepsilon/(B - A)$ and density function $f \in \mathfrak{D}_{\gamma, N}$, we have a density vector $p \in \mathfrak{W}_{\varepsilon, N}$ such that*

$$\left| \int_{z \in [A, B]} h(x, z) \, f(z)\mathrm{d}z - \sum_{k=0}^{K-1} h\left(x, m_k\right) \, p_k \right| \leqslant (M + H\tilde{M})(B - A)^3/(24K^2).$$

## 5.3 Consistency of the uncertainty set

Given $\varepsilon$ and $\delta$, we consider in this section the number of samples $N$ that we need to ensure that the unknown probability density is in the uncertainty set $\mathfrak{D}_{\varepsilon, N}$ with probability $1 - \delta$. The consistency guarantee for the univariate histogram uncertainty set follows as a corollary of the following univariate Dvoretzky-Kiefer-Wolfowitz Inequality.

**Theorem 5.3** ([16]). *Let $\widehat{F}_{N, k}$ denote the distribution function associated with the probabilities $\widehat{p}_{N, K}$, and $F^*$ the distribution function associated with the density function $f^*$. If $F^*$ is continuous, then $\mathbb{P}(\|F^* - \widehat{F}_{N, K}\|_\infty > \varepsilon) \leqslant 2 \exp(-2\varepsilon^2/N)$.*

**Corollary 5.4.** *Let $p^*$ denotes the histogram density vector of $\xi$ induced by the true density $f^*$. As $N \to \infty$, we have $\mathbb{P}(p^* \in \mathfrak{W}_{\varepsilon, N}) \geqslant 1 - 2 \exp(-2\varepsilon^2/N)$.*

*Remark* 2. Provided that the density $f^*$ is Lipchitz continuous, it follows that the optimal value of (A1) converges to the optimal value without uncertainty as the size $\varepsilon$ of the uncertainty set tend to zero and the number of sample $N$ tends to infinity.

# 6 Application to water network optimization

In this section, we consider a problem of optimal operation of a water distribution network (WDN). Let $G = (V, E)$ denote a graph, *i.e.*, $V$ is the set of nodes and $E$ the set of pipes connecting the nodes in a WDN. Let $w_i$ denote the pressure, $e_i$ the elevation, and $\xi^i$ the demand at node $i \in V$, $q_{i,j}$ the flow from $i$ to $j$, and $\ell_{i,j}$ the loss caused by friction in case of flow from $i$ to $j$ for $(i, j) \in E$. Our objective is to minimize the overall pressure at selected critical points $V_1 \subset V$ in the WDN by optimally setting a number of pressure reducing valves (PRVs) located on certain pipes in the network while adhering to the conservations laws for flow and pressure:

$$\min_{(w, q) \in X} h(w, q, \xi), \quad \text{where} \tag{8}$$

$$h(w, q, \xi) := \sum_{i \in V_1} w_i + \sigma \sum_{j \in V} \left( \xi^j - \sum_{k \neq j} q_{k,j} + \sum_{l \neq j} q_{j,l} \right)^2,$$

$$X := \{(w, q) \in \mathbb{R}^{|N|+2|E|} \mid w_{\min} \leqslant w_i \leqslant w_{\max},$$
$$q_{\min} \leqslant q_{i,j} \leqslant q_{\max},$$
$$q_{i,j} \left( w_j + e_j - w_i - e_i + \ell_{i,j}(q_{i,j}) \right) \leqslant 0,$$
$$w_j + e_j - w_i - e_i + \ell_{i,j}(q_{i,j}) \geqslant 0, \quad \forall (i,j) \}.$$

We assume that $\ell_{i,j}$ is a quadratic function in $q_{i,j}$. The PRV sets the pressure $w_i$ at the node $i$. The derivation of (8) and a detailed description of the problem appear in [17]. Thus, $h \in \mathbb{R}[w, q, \xi]$ and $X$ is a basic closed semialgebraic set. For a fixed vector of demands $\xi = (\xi^1, \dots, \xi^{|V|})$, (8) falls into the class (1). In real-world water networks, the demand $\xi$ is uncertain. Given are ranges for the possible realization of nodal demands, *i.e.*, the support of $\xi$ is given by $S := \{\tilde{z} \in \mathbb{R}^{|N|} \mid z_i^{\min} \leqslant \tilde{z}_i \leqslant z_i^{\max}\}$. Moreover, we assume that samples $\xi_1, \dots, \xi_N$ of $\xi$ are given and that they corresponds to sensors measurements. Therefore, the distributionally robust counterpart of (8) is of the form of ADRO (6).

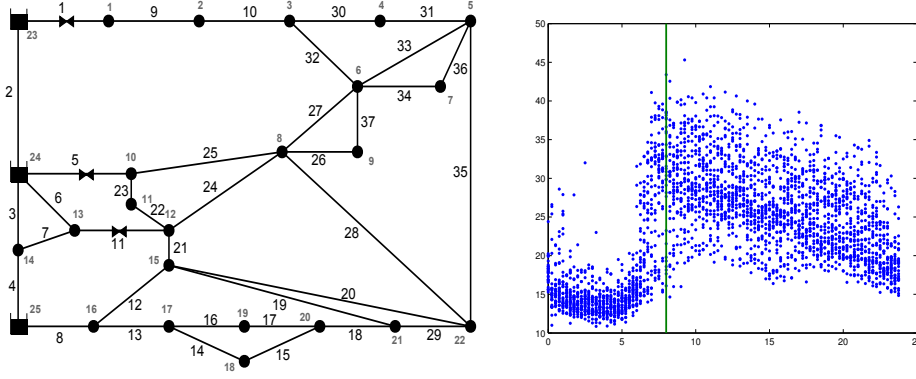

Figure 1: (a) 25 node network with PRVs on pipes 1, 5 and 11. (b) Scatter plot of demand at node 15 over four months overlaid over the 24 hours of a day.

We consider the benchmark WDN with $|V| = 25$ and $|E| = 37$ of [18], which is illustrated in Figure 1 (a). We assign demand values at the nodes of this WDN according to real data collected in an anonymous major city. In our experiment we assume the demands at all nodes, except at node 15, are fixed; for node 15 $N = 120$ samples of daily demands were collected over four months—the dataset is shown in Figure 1 (b). Node 15 has been selected because it is one of the largest consumers and has a demand profile with the largest variation.

First, we consider the uncertainty set $\mathfrak{W}_{\varepsilon, N}$ constructed from a histogram estimation with $K = 5$ bins. We consider, (a) the deterministic problem (8) with three values $\xi_{\min} := \min_i \xi_i^{15}$, $\bar{\xi} := \frac{1}{N} \sum_i \xi_i^{15}$ and $\xi_{\max} := \max_i \xi_i^{15}$ as the demand at node 15, (b) the distributionally robust counterpart (A1) with $\epsilon = 0.2$ and $\sigma = 1$, and (c) the classical robust formulation of (8) with an uncertainty range $[\xi_{\min}, \xi_{\max}]$ without any distributional assumption, *i.e.*, the problem $\min_{(w,q) \in X} \max_{\xi^{15} \in [\xi_{\min}, \xi_{\max}]} h(w, q, \xi^{15})$ which is equivalent to

$$\min_{(w,q) \in X} \max \left( h(w, q, \xi_{\min}), \; h(w, q, \xi_{\max}) \right) \tag{9}$$

since $\left( \xi^{15} - \sum_{k \neq 15} q_{k,15} + \sum_{l \neq 15} q_{15,l} \right)^2$ in (8) is convex quadratic in $\xi^{15}$ attains its maximum at the boundary of $[\xi_{\min}, \xi_{\max}]$. We solve (9) by solving the two polynomial optimization problems.

All three cases (a)–(c), are polynomial optimization problems which we solve by first applying the sparse SDP relaxation of first order [19] with SDPA [20] as the SDP solver, and then applying IPOPT [21] with the SparsePOP solution as starting point. Computations on single blade server with 100GB (total, 80 GB free) of RAM and a processor speed of 3.5GHz. Total computation time is denoted as $t_C$.

| $\xi^{15}$ | $t_C$ | optimal setting | $\sum_{i \in V_1} w_i$ |
|---|---|---|---|
| $\xi_{\min}$ | 738 | $(15.0, 15.7, 15.9)$ | 46.7 |
| $\bar{\xi}$ | 868 | $(15.0, 15.5, 15.6)$ | 46.1 |
| $\xi_{\max}$ | 624 | $(15.0, 15.4, 15.5)$ | 45.9 |

Table 1: Results for non-robust case (a).

| Problem | $t_C$ | optimal setting | objective | $\sum w_i$ |
|---|---|---|---|---|
| DRO (b) | 1315 | $(15.0, 15.5, 15.7)$ | $6.62 \times 10^5$ | 46.2 |
| RO (c) | 1460 | $(15.0, 16.9, 17.3)$ | $1.54 \times 10^6$ | 49.2 |

Table 2: Results for DRO case (b) and classical robust case (c).

The results for the deterministic case (a) show that the optimal setting and the overall pressure sum $\sum_{i \in V_1} w_i$ differ even when the demand at only one node changes, as reported in Table 1.

Comparing the distributionally robust (b) and robust (c) optimal solution for the optimal PRV setting problem, we observe, that the objective value of the distributionally robust counterpart is substantially smaller than the robust one. Thus, the distributionally robust solution is less conservative than the robust solution. Moreover, the distributionally robust setting is very close to the average case deterministic solution $\bar{\xi}$ - but it does not coincide. It seems to hedge the solution against the worst case realization for the demand, given by the scenario $\xi = \xi_{\min}$, which results in the highest pressure profile. Moreover, note that solving the distributionally robust (and robust ) counterpart requires the same order of magnitude in computational time as the deterministic problem. That may be due to the fact that both the deterministic and the robust problems are hard polynomial optimization problems.

## 7 Discussion

We introduced a notion of distributional robustness for polynomial optimization problems. The distributional uncertainty sets based on statistical estimates for the probability density functions have the advantage that they are data-driven and consistent with the data for increasing sample-size. Moreover, they give solutions that are less conservative than classical robust optimization with valued-based uncertainty sets. We have shown that these distributional robust counterparts of polynomial optimization problems remain in the same class problems from the perspective of computational complexity. This methodology is promising for a numerous real-world decision problems, where one faces the combined challenge of hard, non-convex models and uncertainty in the input parameters.

We can extend the histogram method of Section 5 to the case of multivariate uncertainty, but it is well-known that the sample-complexity of histogram density-estimation is greater than polynomial density-estimation. An alternative definition of the distributional uncertainty set $\mathfrak{D}_{\varepsilon, N}$ is to allow functions that are not proper density functions by removing some constraints; this gives a trade-off between reduced computational complexity and more conservative solutions.

The solution method of SDP relaxations comes without any finite-time guarantees. Although such guarantees are hard to come by in general, an open problem is to identify special cases that give insight into the rate of convergence of this method.

**Acknowledgments**

J. Y. Yu was supported in part by the EU FP7 project INSIGHT under grant 318225.

## Footnotes

[1] The function $\Phi(d)$ quantifies the error due to estimation with in a basis of polynomials with finite degree $d$.

[2]Since $S$ is an interval, the assumption is trivially satisfied for $S$.

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
