[Supplementary Material · nips13-full.pdf]

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

# A Proofs

*Proof of Theorem 4.1.* (i): The inner max problem of (5) is equivalent to

$$
\begin{aligned}
\max_{f\in\mathbb{R}[z]_d} \quad & \sum_\alpha x^\alpha \int_S h_\alpha(z)f(z)dz \\
\text{s.t.} \quad & \int_S (f(z) - f^N(z))^2 dz \leqslant \epsilon, \\
& \int_S f(z)\,dz = 1.
\end{aligned}
\tag{10}
$$

For a fixed $x \in X$, (10) can be rewritten as the convex QCQP

$$
\min_{f\in\mathbb{R}[z]_d} \; -c(x)^T f
\tag{11}
$$
$$
\text{s.t. } f^T M f - 2b^T f + b_0 - \epsilon \leqslant 0, v^T f = 1,
$$

where $s(d) := \binom{n+d}{d}$, $c(x) \in \mathbb{R}[x]^{s(d)}$ with $c(x)_\alpha := \int_S h(x,z)\,z^\alpha dz$, $b, d \in \mathbb{R}^{s(d)}$ with $b_\alpha := \int_S z^\alpha f^N(z)dz$ and $v_\alpha := \int_S z^\alpha dz$, $M \in \mathbb{S}_{++}^{s(d)}$ if $\int_S dz > 0$, and $b_0 := \int_S (f^N(z))^2 dz$. Since $f^N \in \mathfrak{D}_{d,\varepsilon,N}$ ($\epsilon > 0$), Slater's condition is satisfied and the Lagrangian dual of (11) has the same optimum. Thus, (5) is equivalent to

$$
\begin{aligned}
\min \quad & -\tfrac{1}{4\lambda}(-c(x) + 2\lambda\,b - \mu\,v)^T M^{-1}(-c(x) + 2\lambda\,b - \mu\,v) \\
& + b_0 - \epsilon - \mu, \\
\text{s.t.} \quad & x \in X, \lambda \geqslant 0, \mu \in \mathbb{R}.
\end{aligned}
\tag{12}
$$

Since $h \in \mathbb{R}[x,z]$, $\widehat{f}_N \in \mathbb{R}[z]$, and $X$ and $S$ are basic closed semi-algebraic sets, (12) is a POP of dimension $m+2$. Since $V^\star \in \mathbb{R}$ and (12) equivalent to (5), for any optimal solution $(x^\star, \lambda^\star, \mu^\star)$ of (12) holds $\lambda^\star, |\mu^\star| \leqslant M$ for some $M > 0$. Thus, the feasible set of (12) can be restricted to set for which Assumption 4.1 holds, and (i) follows from [1, Theorem 1].
(ii): Since (5) has a unique minimizer, the first part $x^\star$ of any optimal solution of (12) unique as well. Thus (ii) follows from (i) and [22, Corollary 13]. $\qquad\square$

*Proof of Corollary 4.3.* First, observe that by assumption, the square-integrable density function $f^*$ has an $L_2$-convergent expansion in Legendre polynomials, *i.e.*, $\lim_{d\to\infty}\|f^* - \sum_{\alpha:|\alpha|\leqslant d} c_\alpha \pi_\alpha\|_2 = 0$. Moreover, by the definitions of $\widehat{f}_N$ and $c_\alpha$, we have $\mathbb{E}\widehat{f}_N = \sum_{\alpha:|\alpha|\leqslant d} c_\alpha \pi_\alpha$. Hence, there exists a function $\Phi$ such that $\Phi(d) \searrow 0$ as $d \to \infty$ and

$$
\left\| f^* - \mathbb{E}\widehat{f}_N \right\|_2 = \left\| f^* - \sum_{\alpha:|\alpha|\leqslant d} c_\alpha \pi_\alpha \right\|_2 \leqslant \Phi(d).
\tag{13}
$$

Observe that by the Triangle Inequality and Chebyshev's Inequality, we have

$$
\begin{aligned}
\mathbb{P}(f^* \notin \mathfrak{D}_{d,\varepsilon,N}) &= \mathbb{P}\left( \|f^* - \widehat{f}_N\|_2 > \varepsilon \right) \\
&\leqslant \mathbb{P}\left( \|\widehat{f}_N - \mathbb{E}\widehat{f}_N\|_2 + \|\mathbb{E}\widehat{f}_N - f^*\|_2 > \varepsilon \right) \\
\text{(by (13))} &\leqslant \mathbb{P}\left( \|\widehat{f}_N - \mathbb{E}\widehat{f}_N\|_2 > \varepsilon - \Phi(d) \right) \\
\text{(by Chebyshev)} &\leqslant \mathbb{E}\|\widehat{f}_N - \mathbb{E}\widehat{f}_N\|_2^2 / (\varepsilon - \Phi(d))^2 \\
&\leqslant \frac{\mathbb{E}\|\widehat{f}_N - f^*\|_2^2 + \|f^* - \mathbb{E}\widehat{f}_N\|_2^2}{(\varepsilon - \Phi(d))^2},
\end{aligned}
$$

from which follows the claim by (13) and Theorem 4.2. $\qquad\square$

*Proof of Theorem 5.1.* (i): The inner maximization of (7) can be written as

$$
\begin{aligned}
\text{(M1)} \quad \max_{p\in\mathbb{R}^K} \quad & \sum_{k=0}^{K-1} h\,(x,m_k)\,p_k \\
\text{s.t.} \quad & p_k - \widehat{p}_{N,K}(k) \leqslant \varepsilon,\ k = 0,\ldots,K-1; \\
& \widehat{p}_{N,K}(k) - p_k \leqslant \varepsilon,\ k = 0,\ldots,K-1; \\
& \sum_{i=1}^K p_i \leqslant 1; p_j \geqslant 0,\ j = 0,\ldots,K-1.
\end{aligned}
$$

By taking the dual of the inner maximization (M1) and by the Strong Duality for LPs—the primal and dual have the same optimal value, and (7) can be written as

$$\text{(A1)} \quad \min_{\substack{x \in X \\ y \in \mathbb{R}^{2K+1}}} \quad \sum_{k=0}^{K-1} y_k(\varepsilon + \widehat{p}_{N,K}(k))$$

$$+ \sum_{\ell=K}^{2K-1} y_\ell(\varepsilon - \widehat{p}_{N,K}(\ell - K)) + y_{2K}$$

$$\text{s.t.} \quad y_k - y_{K+k} + y_{2K} \geqslant h(x, m_k),$$
$$k = 0, \ldots, K-1,$$
$$y_j \geqslant 0, \quad j = 0, \ldots, 2K.$$

(A1) is a POP of dimension $n + 2K + 1$, its degree coincides with the degree of $h$. Since $W^\star \in \mathbb{R}$ and (A1) equivalent to (7), for any optimal solution $(x^\star, y^\star)$ of (A1) holds $y_j^\star \leqslant M$ for some $M > 0$ for all components $j \in \{0, \ldots, K-1\}$. Thus, the feasible set of (A1) can be restricted to a basic, closed semi-algebraic set for which Assumption 4.1 holds, and (i) follows from [1, Theorem 1]. The proof of (ii) is similar to Theorem 4.1 (ii). □

*Proof of Theorem 5.2.* Consider a fixed $x$. Given $f \in \mathfrak{D}_{\gamma,N}$, we construct the density vector $p \in \Delta^K$: $p_k = \int_{a \in u_k} f(a)\mathrm{d}a$, for all $k$. Observe that, since $f \in \mathfrak{D}_{\gamma,N}$, we have

$$|p_k - \widehat{p}_{N,K}(k)| = \left| \int_{u_k} f(a)\mathrm{d}a - \widehat{f}_{N,K}(a)\mathrm{d}a \right|$$
$$\leqslant |u_k| \|f - \widehat{f}_{N,K}\|_\infty \leqslant \frac{B-A}{K} \|f - \widehat{f}_N\|_\infty \leqslant \frac{B-A}{K}\gamma \leqslant \varepsilon,$$

so that $p \in \mathfrak{W}_{\varepsilon,N}$.

Observe that by the definition of $\mathfrak{D}_{\gamma,N}$, for $f \in \mathfrak{D}_{\gamma,N}$, we have $f''(z) \leqslant \tilde{M} < \infty$ for all $z \in [A, B]$. Moreover, since $f$ and $h$ are polynomials, we have $g_x''(z) \leqslant M < \infty$ for all $z \in [A, B]$ and all $x \in X$. Hence, the assumptions of the Riemann Approximation Theorem are satisfied, and we have

$$\left| \int_{z \in S} h(x, z)\, f(z)\mathrm{d}z - \sum_{k=0}^{K-1} h(x, m_k)\, f(m_k)\, |u_k| \right|$$
$$\leqslant M(B-A)^3/(24K^2),$$

where $m_k$ midpoint of the interval $u_k$. Next, observe that

$$\left| \sum_{k=0}^{K-1} h(x, m_k)\, p_k - \sum_{k=0}^{K-1} h(x, m_k)\, f(m_k)\, |u_k| \right|$$
$$\leqslant \left| \sum_{k=0}^{K-1} h(x, m_k)\, (\int_{a \in u_k} f(a)\, \mathrm{d}a - f(m_k)\, |u_k|) \right|$$
$$\leqslant H \left| \sum_{k=0}^{K-1} (\int_{a \in u_k} f(a)\, \mathrm{d}a - f(m_k)\, |u_k|) \right|$$
$$\leqslant H\tilde{M}(B-A)^3/(24K^2),$$

where the last inequality follows again from the Riemann Approximation Theorem. The claim follows for these two inequality by applying the Triangle Inequality. □

*Proof of Corollary 5.4.* The claim follows from [16, Corollary 1] by observing that $\|p^* - \widehat{p}_{N,K}\|_\infty \leqslant \|F^* - \widehat{F}_{N,K}\|_\infty$. □