[Reviews · NeurIPS 2013]

Submitted by Assigned_Reviewer_3

The authors consider robust polynomial optimization where the uncertainty parameter is a probability distribution estimated from data samples. The solution is shown to be the limit of a sequence of SDP relaxations and finite-sample consistency guarantees are given. Numerical results on a water network problem are provided.


Quality: The paper is technically sound. The proofs are provided in the supplementary material.

minor comments:
1) A more detailed explanation (example) on constructing distributionally uncertainty sets from sampled information may be helpful.

2) "Table 6" on page 7 should be "Table 2". It seems to be hard to justify the advantage of the proposed robust approach from the given example.

Clarity: The paper is well-organized.

Originality: Forming distributionally robust polynomial optimization into polynomial optimization problems and constructing distributionally uncertainty sets from sampled information sound interesting. The asymptotic guarantees in Theorems 4.1 and 5.1 follow from the references [1] and [22].

Significance: Polynomial optimization arises from many fields. A practically useful robust polynomial optimization approach with certain theoretical guarantees is important.
Summary: Robust polynomial optimization using probability distribution estimated from data samples is interesting. The asymptotic guarantees seem to be extensions of the results in the literature.

Submitted by Assigned_Reviewer_4

The authors considered robust optimization for polynomial optimization problems where the uncertainty set is a set of possible distributions of the parameter. In specific, this set is a ball around a density function estimated from data samples. The authors showed that this distributionally robust optimization formulation can be reduced to a polynomial optimization problem, hence computationally the robust counterpart is of the same hardness as the nominal (non-robust) problem, and can be solved using a tower of SDP known in literature. The authors also provide finite-sample guarantees for estimating the uncertaity set from data. Finally, they applied their methods to a water network problem.

This seems to be an interesting paper. Distributionally robust optimization is now topical in operations research. Most previous literature is restricted to the case that the distribution is characterized by (first and second) moments, which has the apparent disadvantage that consistency is impossible regardless of the number of samples obtained. In contrast, the proposed approach overcomes this difficulty, as when more samples are obtained, we can shrink the uncertainty set until it becomes a singleton, and hence achieving consistency. It is also nice to see that, at least for the polynomial optimization problems, the resulting robust counterpart does not incur additional computational difficulty.

Some minor points:

1. Assumption 4.1 is a bit confusing. In particular, I am not sure how the sentence "and there exist u\in R[x]... compact." is related to the set X and S.
2. In line 194, "\hat{S} satisfies Assumption 4.1". Why?
3. L236, "We consider the multi-dimensional case in the next section" appears to be wrong.
4. L375 "Table 6". There is no table 6 in the submission.
5. L379 "Un-biased". I am not sure why the uncertainty set are unbiased (in fact I am even sure what does it mean that a set is unbiased).
6. L148, "A notion of distributional uncertainty set has also been studied in the setting of Markov decision problems [14]". I think [14] does not consider distributional uncertainty. To my knowledge, distributional uncertainty in MDP is investigated in Xu and Mannor (2012).

Ref:
H. Xu, S. Mannor, "Distributionally Robust Markov Decision Processes", Mathematics of Operations Research, 37(2), pp288-300, 2012.
Summary: This seems to be an interesting paper. Distributionally robust optimization is now topical in operations research. Most previous literature is restricted to the case that the distribution is characterized by (first and second) moments, which has the apparent disadvantage that consistency is impossible regardless of the number of samples obtained. In contrast, the proposed approach overcomes this difficulty, as when more samples are obtained, we can shrink the uncertainty set until it becomes a singleton, and hence achieving consistency. It is also nice to see that, at least for the polynomial optimization problems, the resulting robust counterpart does not incur additional computational difficulty.

I have taken into consideration the rebuttal.

Submitted by Assigned_Reviewer_5

The paper introduces the notion of data driven robust optimisation(DRO) to solve optimization
problems with inherent uncertainty.
In particular it investigates a min-max problem over the expectation of a polynomial function.
The max is taken over all allowed probability densities and the min is over the decision
variable. The results appear to be correct and could be of significant interest to robust optimization
community. It would have been nice if the paper could discuss a relevant ML problem in the context of
DRO.

Quality: The paper scores high in mathematical rigour.
.

Clarity: The paper is clearly written and the results are properly stated.

Originality: The results are novel.

Significance: The results in this paper could be of significant interest to Operations Research community.
Summary: The main results of this paper may not be of interest for a ML community.
Author Feedback

Author rebuttal: 